# Proteomic Analysis of Human Immune Responses to Live-Attenuated Tularemia Vaccine

**DOI:** 10.3390/vaccines8030413

**Published:** 2020-07-24

**Authors:** Yie-Hwa Chang, Duc M. Duong, Johannes B. Goll, David C. Wood, Travis L. Jensen, Luming Yin, Casey E. Gelber, Nicholas T. Seyfried, Evan Anderson, Muktha S. Natrajan, Nadine Rouphael, Robert A. Johnson, Patrick Sanz, Mark J. Mulligan, Daniel F. Hoft

**Affiliations:** 1Edward A. Doisy Department of Biochemistry and Molecular Biology, Saint Louis University Medical School, Saint Louis, MO 63104, USA; david.wood@health.slu.edu; 2Department of Biochemistry, Emory School of Medicine, Atlanta, GA 30322, USA; dduong@emory.edu (D.M.D.); lyin2@emory.edu (L.Y.); nseyfri@emory.edu (N.T.S.); 3The Emmes Company, Rockville, MD 20850, USA; jgoll@emmes.com (J.B.G.); tjensen@emmes.com (T.L.J.); cgelber@emmes.com (C.E.G.); 4Department of Pediatrics, Emory University School of Medicine and Children’s Healthcare of Atlanta, Atlanta, GA, 30322, USA; evanderson@emory.edu; 5Division of Infectious Diseases, Department of Medicine, Emory University School of Medicine, Atlanta, GA 30322, USA; muktha.natrajan@emory.edu (M.S.N.); nroupha@emory.edu (N.R.); 6Emory Vaccine Center, Emory University, Atlanta, GA 30322, USA; 7Biomedical Advanced Research and Development Authority, U. S. Department of Health and Human Services, Washington, DC 20201, USA; Robert.Johnson@hhs.gov; 8Division of Microbiology and Infectious Diseases, National Institute of Allergy and Infectious Diseases, National Institutes of Health, Rockville, MD 20892, USA; patrick.sanz@nih.gov; 9Division of Infectious Diseases and Immunology, Department of Medicine, and New York University (NYU) Langone Vaccine Center, NYU School of Medicine, New York, NY 10016, USA; mark.mulligan@nyulangone.org; 10Department of Internal Medicine, Saint Louis University Medical School, Saint Louis, MO 63104, USA; Daniel.hoft@health.slu.edu

**Keywords:** proteomics, human immune response, tularemia vaccine, 2D-DIGE, LC-MS/MS, antigen processing, inflammation, phagocytosis, biomarkers

## Abstract

*Francisella tularensis* (*F. tularensis*) is an intracellular pathogen that causes a potentially debilitating febrile illness known as tularemia. *F. tularensis* can be spread by aerosol transmission and cause fatal pneumonic tularemia. If untreated, mortality rates can be as high as 30%. To study the host responses to a live-attenuated tularemia vaccine, peripheral blood mononuclear cell (PBMC) samples were assayed from 10 subjects collected pre- and post-vaccination, using both the 2D-DIGE/MALDI-MS/MS and LC-MS/MS approaches. Protein expression related to antigen processing and presentation, inflammation (PPARγ nuclear receptor), phagocytosis, and gram-negative bacterial infection was enriched at Day 7 and/or Day 14. Protein candidates that could be used to predict human immune responses were identified by evaluating the correlation between proteome changes and humoral and cellular immune responses. Consistent with the proteomics data, parallel transcriptomics data showed that MHC class I and class II-related signals important for protein processing and antigen presentation were up-regulated, further confirming the proteomic results. These findings provide new biological insights that can be built upon in future clinical studies, using live attenuated strains as immunogens, including their potential use as surrogates of protection.

## 1. Introduction

*Francisella tularensis* (*F. tularensis)*, a gram-negative coccobacillus, is an intracellular pathogen of humans and animals that causes a potentially debilitating febrile illness known as tularemia [1,2,3]. *F. tularensis* infection occurs after exposure to infected wildlife species (including rodents, hares and rabbits), or through bites from infected ticks and deer flies, and causes disease of varying severity dependent upon the portal of entry, the infectious dose, and the subspecies (biovar) of the infecting strain [1,2,3,4]. Two biovars have been described: type A, *F. tularensis* biovar *tularensis* and type B, *F. tularensis* biovar *holarctica*. The type A *F. tularensis* subspecies is the most infectious biovar (ID_50_ < 10 cfu), which is responsible for most cases of tularemia in North America. If untreated, this subspecies has mortality rates approaching 30% [3,4,5,6]. *F. tularensis* can be spread by aerosol transmission to cause fatal pneumonic tularemia and is therefore classified as a Tier 1 select agent of bioterrorism, indicating its status as one of the most likely pathogens to be deliberately used in a bioterrorist attack [7,8]. Japan, the U.S.A., and the U.S.S.R. have previously stockpiled *F. tularensis* as a potential bioweapon. Given the dangers of *F. tularensis* infection, more research on both the vaccination and molecular-level effects of *F. tularensis* on human health is needed [8,9,10].

The United States Army Medical Research Institute of Infectious Diseases—LVS (USAMRIID-LVS) live-attenuated vaccine has been used under an investigational new drug (IND) application for decades [11]. However, supply is limited and aging. Thus, the U.S. Department of Defense contracted with Dynport Vaccine Company (DVC) to produce new lots of live vaccine strain (LVS) using current good manufacturing practices (cGMP). It is worth noting that the USAMRIID and DVC-LVS are the same vaccine, that is filed under the same IND application. A phase 1 trial of escalating doses of the new DVC-LVS lot (lot 17) administered to 70 subjects concluded that vaccine delivery by scarification was safe, tolerable, and produced superior antibody responses relative to subcutaneous delivery [12,13]. The data from the phase 2 trial directly compared the new DVC-LVS lot (lot 20) to USAMRIID-LVS in 228 subjects defining the kinetics of antibody responses, comparing injection site reactions following scarification, and correlating antibody responses with take [14]. The availability of pre- and post-vaccination samples from this trial offers a unique opportunity to better characterize host immune responses to this live-attenuated vaccine, to improve vaccine development. As a first step towards this goal, PBMC gene expression responses were previously assessed based on microarrays, following vaccination with two tularemia vaccines [14,15]. 

In this study, the goal was to comprehensively study protein changes that occur in response to Tularemia DVC-LVS vaccination. Proteomics encompasses the large-scale quantification and identification of the entire set of cellular proteins, as well as the characterization of their modifications, functions, and interactions. It is a powerful tool for studying the host responses to infection and immunization. With the advent of mass spectrometry, microcapillary chromatography, and genome-assisted data analysis, the number, speed, and sensitivity of the proteins and post-translational modifications identified in samples has increased significantly [16]. More importantly, the development of two-dimensional difference gel electrophoresis (2D-DIGE) technology significantly improved reproducibility and accuracy in quantifying proteins expressed within two sets of samples [17,18]. However, one limitation of this technology is its relatively low throughput. To overcome this problem, different quantitative LC-MS/MS methods have been developed, with various advantages and disadvantages. They include spectral-counting, stable isotope labeling by amino acids in cell culture (SILAC), iTRAQ, isotope-coded affinity tag (ICAT) and tandem mass tag (TMT), which have allowed for both absolute and relative quantification of proteins in complex samples [19,20]. Protein microarrays have been developed to track the interactions and activities of pre-determined numbers of proteins in parallel [21,22].

To study the host responses post-vaccination with a live-attenuated tularemia vaccine, 2D-DIGE and LC-MS/MS analyses were performed using PBMC cell lysates from 10 subjects collected pre- vaccination (Day 0) and post-vaccination (Days 7 and 14). PBMC samples were assayed by two different laboratories using two different proteomics protocols. RNA-Seq based transcriptomics, metabolomics and lipidomics experiments on the same cohort and timepoints which are cross-referenced in the discussion section (23 and unpublished data) were also performed. Here, we report the novel findings regarding the activation of PBMC protein expression related to antigen processing and presentation, inflammation (PPARγ receptor), phagocytosis, and gram-negative bacterial infection following DVC-LVS vaccination.

## 2. Materials and Methods 

### 2.1. Study Design

The tularemia vaccine clinical trial (ClinicalTrials.gov identifier NCT01150695) was designed as a double-blind, randomized study of a single, undiluted dose of the *F. tularensis* DVC-LVS lot produced by DynPort Vaccine Company via scarification, versus a single, undiluted dose of the USAMRIID-LVS vaccine via scarification (multiple puncture technique) on Day 0 [12,13,14]. Both vaccines were administered in the ulnar aspect of the volar surface (palm side) of the forearm midway between the wrist and the elbow [12,13,14]. Blood samples were taken from enrolled subjects at multiple timepoints. This proteomics pilot study included 10 subjects who had agreed to future-use and had available samples from the DVC-LVS vaccine group, with 30 total samples collected from subjects at Days 0 (prior to vaccination), 7, and 14. In one case, 2D-DIGE methodology for all samples and MALDI-QIT-TOF mass spectrometry to identify proteins within differentially abundant (DA) gel spots were sequentially applied. In parallel, a LC-MS/MS approach was carried out on PBMC lysates after trypsin digestion, followed by protein identification and relative abundance calculation using a label-free quantification method to determine differential protein abundance. Samples from the same subjects were also studied by microarrays [15], RNAseq (unpublished data) and metabolomics [23], as well as lipidomics [23], to determine changes in gene expression, lipids, and metabolites (results of these other omic platforms are reported separately).

### 2.2. Protein Spike-in Controls 

A spike-in control protein mixture of bovine beta-lactoglobulin (Sigma L-5137), horse myoglobin (Sigma M-9267), and bovine ribonuclease A (Sigma R-6513) was prepared in 8 M urea. Stock solutions of each protein were prepared in Milli-Q grade water, quantified by UV absorbance spectrum, and combined to make 50 mL of 8 M urea solution containing 300 ng/mL of each protein.

### 2.3. Shared Protein Sequence Database and Protein Families 

The combined human subset (organism restricted to *Homo sapiens*), defined by both the UniProtKB/Swiss-Prot and UniPro- tKB/TrEMBL Release 2016-03 databases, was used as a reference for proteomics searches. The *CD-HIT* software (Version 4.0 beta, La Jolla, USA) was used to derive protein clusters at a 50% protein sequence identity level (henceforth, 50% CD-HIT protein clusters are referred to as protein families). Prior to proteomic searching, identical protein sequences were collapsed, and the three spike-in control protein sequences were added to the protein sequence database (UniProt Accession P02754: Beta-lactoglobulin (Bovine), UniProt Accession P68082: Myoglobin (Horse), and UniProt Accession P61823: Ribonuclease pancreatic (Bovine)).

### 2.4. LC-MS/MS Proteomics Experiment and Data Processing

One mL of ice-cold PBS was added to each sample and the samples were centrifuged for 10 min at 400× *g*. Afterwards, 1.5 mL of supernatant was removed and discarded. An additional 500 µL of ice-cold PBS was added and the samples were spun again at 1000× *g* for 10 min. The supernatant was removed, and the pellet was resuspended in 300 µL of urea lysis buffer (8M Urea spiked with 3 protein spike-in controls), including a 3 µL (100× stock) HALT protease and phosphatase inhibitor cocktail (Pierce). The entire mixture was then sonicated (Sonic Dismembrator, Fisher Scientific) 3 times for 5 s, with 15 s intervals of rest at 30% amplitude to disrupt nucleic acids and was subsequently vortexed. Protein concentration was determined by the bicinchoninic acid (BCA) method, and samples were frozen in aliquots at −80 °C. Protein homogenates (100 μg) were diluted with 50 mM NH_4_HCO_3_ to a final concentration of less than 2M urea and then treated with 1 mM dithiothreitol (DTT) at 25 °C for 30 min, followed by 5 mM iodoacetamide (IAA) at 25 °C for 30 min in the dark. Protein was digested with 1:100 (*w/w*) lysyl endopeptidase (Wako Chemicals USA, Inc.) at 25 °C for 2 h and further digested overnight with 1:50 (*w/w*) sequencing grade trypsin (Promega Corporation) at 25 °C [17]. Resulting peptides were desalted with a Sep-Pak C18 column (Waters) and dried under vacuum.

Dried peptides were resuspended in 100 µL of loading buffer (0.1% formic acid, 0.03% trifluoroacetic acid, 1% acetonitrile). Peptide mixtures (2 µL) were separated on a self-packed C18 (1.9 μm Dr. Maisch, Germany) fused silica column (25 cm × 75 µM internal diameter (ID); New Objective, Woburn, MA) by a Dionex Ultimate 3000 RSLCNano and monitored on a Fusion mass spectrometer (ThermoFisher Scientific, San Jose, CA). Elution was performed over a 140 min gradient, at a rate of 300 nl/min, with buffer B ranging from 3% to 80% (buffer A: 0.1% formic acid in water, buffer B: 0.1% formic in acetonitrile). The mass spectrometer cycle was programmed to collect at the top speed for 3-s cycles. The MS scans (400–1600 *m/z* range, 200,000 AGC, 50 ms maximum ion time) were collected at a resolution of 120,000 at *m/z* 200 in profile mode and the HCD MS/MS spectra (2 *m/z* isolation width, 30% collision energy, 10,000 AGC target, 35 ms maximum ion time) were detected in the ion trap. Dynamic exclusion was set to exclude previous sequenced precursor ions for 20 s within a 10 ppm window. Precursor ions with +1, and +8 or higher charge states were excluded from sequencing. RAW data for samples were analyzed using MaxQuant v1.5.3.30, with Thermo Foundation for RAW file reading capability. The search engine Andromeda, integrated into MaxQuant, was used to build and search a concatenated target-decoy human reference protein database (20,157 target entries plus 245 contaminant proteins from the common Repository of Adventitious Proteins (cRAP) built into MaxQuant). Methionine oxidation (+15.9949 Da), asparagine and glutamine deamidation (+0.9840 Da), and protein N-terminal acetylation (+42.0106 Da) were variable modifications (up to 5 allowed per peptide); cysteine was assigned a fixed carbamidomethyl modification (+57.0215 Da). Only fully tryptic peptides were considered, with up to 2 miscleavages in the database search. A precursor mass tolerance of ±20 ppm was applied prior to mass accuracy calibration and ±4.5 ppm after internal MaxQuant calibration. Cofragmented peptide search was enabled to deconvolute multiplex spectra. The false discovery rate (FDR) for peptide spectral matches, proteins, and site decoy fraction was set to 1 percent. Quantification settings were as follows: re-quantify with a second peak finding attempt after protein identification has completed; match MS1 peaks between runs; a 0.7 min retention time match window was used after an alignment function was found with a 20-min RT search space. Label-free quantification of proteins and normalization was performed using the MaxLFQ algorithm, as implemented in MaxQuant. The quantitation method only considered razor plus unique peptides for protein level quantitation. Protein group signals were filtered to retain groups, for which at least two unique peptides were identified. Zero intensity values were set to missing. The leading protein in a protein group (the protein with the highest number of identified peptides) was used as the representative protein for each group. Information about other proteins in a protein group was retained and integrated when presenting lists of differentially abundant proteins.

### 2.5. 2D-DIGE/MS Proteomics Experiment and Data Processing

PBMC cell pellets were prepared by the controlled thawing of cryopreserved cells in order to maximize viability, which was assessed by trypan blue stain. Cells were counted by hemocytometer and washed twice with PBS buffer, prior to freezing in aliquots of 2 million viable cells per vial. Cell pellets were lysed, proteins precipitated, and total protein quantified using a Pierce 660 nm protein assay. Prior to carrying out DIGE sub-stoichiometric Cy Dye labeling, an equal protein quantity was taken from each lysate to create a pool for normalization of fluorescence intensity for all analytical 2D gels. This pool was labeled with Cy2 dye, and individual samples were labeled with either Cy3 or Cy5 (Lumiprobe, MD), using a dye ratio of 8 pmol per μg protein [17]. The samples were analyzed by 2D gel electrophoresis using pH 3–11 NL IPG strips (GE Healthcare) and Criterion XT 4–12% SDS-PAGE gels (BioRad Laboratories, CA). Fluorescence imaging was carried out using a Typhoon 9410. Raw data files were cropped and filtered using ImageQuant TL 8.1 and analyzed using DeCyder 6.5, to determine protein spot relative abundance for Days 7 or 14 versus Day 0. Spots corresponding to spiked proteins were identified using MALDI-TOF mass spectrometry. Forty-one 2D-DIGE gel spots were linked to 68 reference sequence database entries (UniProt IDs).

### 2.6. Statistical Analysis

#### 2.6.1. Normalization

Median normalization was performed to account for systematic differences in protein signal distributions, by aligning the medians of the 30 *log*2 protein signal distributions, which involved the following steps:(1)for each sample, the median of the *log*2 protein signal distribution was determined.(2)the global median of all 30 sample medians calculated in (1) was obtained.(3)a sample specific scaling factor was then calculated as the difference (*log*2 scale) between the global median obtained in (2) and the sample-specific median obtained in (1).(4)the *log*2 protein signal distribution for each sample was then median normalized by adding the scaling factor (*log*2 scale), determined in (4).

Local regression (LOESS)-based normalization, as implemented in the *affy* R package (Version 1.48.0), was used to correct systematic signal-dependent non-linear bias observed for Cy5, versus Cy3-labeled 2D-DIGE data samples.

#### 2.6.2. Missing Value Imputation and Log Fold Change from Baseline Calculation

Missing observations were imputed using the k-nearest neighbors algorithm implemented in the impute R package (Version 1.44.0). Only proteins/spots with at least 24/30 (80%) non- missing observations were used as input for imputation and downstream analysis. The number of neighbors to be used as part of the imputation step was set to 8. Subject-specific *log*2 protein fold changes from baseline were calculated based on normalized imputed *log*2 signals for each subject and post-vaccination day (Days 7 and 14), by subtracting baseline (Day 0) protein signals from each of the subject’s post-vaccination day signals.

#### 2.6.3. Identification of Differentially Abundant Proteins

Proteins that significantly differed in their response from baseline were identified by using a two-sided permutation paired t-test, comparing post-vaccination (Day x) to baseline (Day 0) protein signals (H0:µ(day_x_ − day0) = 0, H1:µ(day_x_ − day0) ≠ 0; on the log2 scale). Proteins with an individual *p*-value <0.05 and baseline fold change ≥1.2 were considered significantly differentially abundant (DA) proteins.

#### 2.6.4. Pathway Enrichment Analysis

Pathway enrichment analysis was carried out separately for each post-vaccination day, using 5850 known gene sets obtained from the KEGG Pathway (Version 79.0, 07/16/2016) and MSigDB (Version 5.1, 01/19/2016 including MSigDB Reactome Pathways and MSigDB Immunologic Signatures) databases. Prior to pathway enrichment, proteins in the proteomics protein database were mapped to Ensembl Gene IDs (Ensembl release 84, March 2016), using the biomaRt R package (Version 2.26.1), based on their UniProt protein accessions. If a UniProt protein accession mapped to multiple Ensembl Gene IDs, multiple Ensembl Gene IDs were assigned to that protein. Following the mapping step, genes in gene sets without any UniProt protein accession mappings were excluded from the gene set collections. Gene set statistics after filtering are provided in Appendix A.

For each of the filtered gene sets, enrichment was evaluated using the goseq R package (Version 1.12.0), using the hypergeometric distribution to assess statistical significance. To adjust for testing multiple gene sets per category type, the Benjamini–Hochberg procedure was applied to each list. Pathways with an FDR ≤0.1 were considered to be significantly enriched. For significantly enriched KEGG pathways, color-coded KEGG pathway maps were generated (KEGG KGML pathway layout information Version 81.0, 01/06/2017). Node background was color-coded by mean log2 fold change from pre-vaccination (red: up-regulated from baseline, green: down-regulated from baseline). For the 2D-DIGE/MS data, the largest mean log_2_ fold change was used for UniProt IDs with multiple gel spot IDs. If nodes in the pathway referred to multiple genes, the median log2 fold change was used to set the background color of that node (red: up-regulated from baseline, green: down-regulated from baseline). If one of the genes of a multi-gene node was significantly enriched, the node label and border was color-coded (red: up-regulated, green: down-regulated, blue: conflict if one gene was up but another was down-regulated for the same pathway node).

#### 2.6.5. Identification of Protein Responses that Best Predicted Humoral and Cellular Responses

Regularized linear regression models were fit to determine the 2D-DIGE and LC-MS/MS protein log_2_ fold change responses that best predicted peak percent activated CD4^+^ T-cells (CD3^+^CD4^+^CD38^+^HLA^-^DR^+^ cells), peak percent activated CD8^+^ T-cells (CD3^+^CD8^+^CD38^+^HLA^-^DR^+^ cells), and peak microagglutination titer, using the glmnet R package (Version 2.0–13). T-cell and microagglutination data was taken from [14]. T-cell variables were encoded as peak percent activated cells across Days 7, 14, and 28, while tularemia-specific microagglutination was encoded as the log_2_ of the peak titer across Days 14 and 28. Leave-one-out cross validation was used to determine the optimum regularization parameters *α* and *λ*, that minimized the model mean squared error. Models were considered to fit the data well if they achieved an R^2^ of at least 0.7. Correlation networks were for selected proteins were constructed using Pearson correlation (r ≥ 0.4) and visualized using the R igraph package (Version 1.2.2).

### 2.7. Transcriptomics Data

Transcriptomics data from parallel study were used to assess transcriptome-wide gene expression in PBMCs using RNA-Seq for subjects analyzed in this study (GEO Accession GSE149809).

## 3. Results

In this study, PBMC cell lysates from 10 subjects collected at pre-vaccination (Day 0) and post-vaccination (Days 7 and 14) with a live attenuated tularemia vaccine were assayed by two different laboratories. Each laboratory used a different proteomics protocol. On average, 748 gel spots were identified for each 2D-DIGE gel, while 1872 proteins and 1517 protein families were identified in each sample when using LC-MS/MS (Appendix A).

### 3.1. Technical Assessments and Method Comparisons

Using MALDI-TOF based identification, 2D gels were characterized, resulting in 41 DA gel spots linked to 68 UniProt reference sequence database entries, of which 59 were unique. Thirty-five of these were included in the LC-MS/MS imputed analysis dataset. A listing of all gel spot IDs with protein identifications are provided in Appendix A. The agreement of protein abundance for the 35 shared proteins, as measured in log_2_ label-free quantification (LC/MS/MS) and gel spot volume ratios (2D-DIGE), as well as the agreement between log_2_ fold change from baseline, was evaluated. The mean Pearson correlation between 2D-DIGE log_2_ spot volume ratios and the LC-MS/MS log_2_ label-free quantification for these 35 shared proteins was 0.28 (based on 30 samples). The mean Pearson correlation between baseline log_2_ fold changes was 0.10 (based on 20 samples).

Boxplots that contrast spike-in protein (beta-lactoglobulin (bovine), myoglobin (horse), and ribonuclease pancreatic (bovine)) variability across the 30 samples, within and between proteomics methods, are displayed in Appendix A. In the 2D-DIGE analysis, RNaseA could not confidently be assigned to a spot in the master gel. Thus, only beta-lactoglobulin and myoglobin were summarized for the 2D-DIGE data. For the LC/MS/MS analysis, RNaseA was identified, but at a much lower abundance than either beta-lactoglobulin or myoglobin (Appendix A). For each proteomics experiment, the coefficient of variation (CV) and robust median absolute deviation (MAD) were calculated (see *x*-axis labels in Appendix A). To compare the variability more directly between experiments, log_2_ mean-centered protein signals were used. For both experiments, the beta-lactoglobulin (bovine) spike-in protein showed higher variability compared to myoglobin (horse). While beta-lactoglobulin (bovine) was similar in variability (as assessed by MAD) between experiments following normalization, differences were more pronounced for myoglobin (horse). This was also observed when contrasting interquartile ranges for mean-centered protein signals between experiments (Appendix A, bottom right). Overall, the MAD was lower for the 2D-DIGE experiment compared to the LC-MS/MS experiment, with an 8% reduction in MAD for bovine beta- lactoglobulin (0.58 vs. 0.63) and a 33% reduction in MAD for horse myoglobin protein (0.35 vs. 0.52) (Appendix A and Appendix A).

### 3.2. DA Proteins and Enriched Pathways

A permutation paired t-test to identify vaccine-responsive proteins was performed (Appendix A). Overall, the 2D-DIGE experiment yielded 88 gel spot identifications, that showed DA signals at any post-vaccination day (Day 7 or 14) relative to pre-vaccination (Day 0) (Appendix A). Twenty were shared between post-vaccination days. Notably, 60% percent were upregulated from pre-vaccination (Figure 1 and Appendix A). For the LC-MS/MS experiment, 92 proteins were DA compared to pre-vaccination with 10 being shared between post-vaccination days. Moreover, 96% were up regulated from pre-vaccination (Figure 1 and Appendix A). For both experiments, more DA proteins were identified at day 14 compared to day 7. Four of the 35 (11.4%) shared proteins were DA for both methods (Table 1). For these 4 shared DA proteins, LC/MS/MS data tended to have higher fold changes compared to 2D-DIGE for most proteins. The directions of fold changes for all four DA proteins matched between experiments (all were upregulated from pre-vaccination), ranging from 1.3 to 2.2 -fold increase at Days 7 or 14 compared to pre-vaccination (Table 1). These proteins included TPI1, which plays an important role in energy production/glycolysis (source UniProt) and is related to CD8^+^ T cell activation [24,25], as well as 3 antigen presentation-related proteins comprising KIF5B (involved in MHC class II antigen presentation, source UniProt), HSP90AA1 (involved in chaperoning peptide antigens to be loaded onto MHC I molecules, source: KEGG), and CAPZA1 (involved in MHC class II antigen presentation, source UniProt). The strongest fold change following vaccination for both experiments was observed for the HSP90AA1 chaperone at Day 14 (2.2-fold increase for LC-MS/MS data and 2.0-fold increase for 2D-DIGE data) (Table 1).

To evaluate the functional context of DA proteins, pathway enrichment analysis was carried out for each experiment (LC-MS/MS and 2D-DIGE/MS data) and post-vaccination day (Days 7 and 14). Significantly enriched KEGG pathways are provided in Table 2. All enrichment results, including those for MSigDB reactome pathways and MSigDB immunologic signatures, are listed in Appendix A. KEGG Pathway enrichment analysis of the 2D-DIGE data showed that the antigen presentation and processing pathway was enriched in DA upregulated proteins on Day 7 and Day 14 (Figure 2). The protein processing in the endoplasmic reticulum pathway was enriched in DA upregulated proteins on Day 14 (Figure 3). Stronger upregulation on Day 14 indicated that there was an increase in antigen presentation response over time. The phagosome pathway was also enriched in DA upregulated proteins on Day 14 (Appendix A). For the LC-MS/MS data, the ribosome pathway was enriched in upregulated DA proteins on both Days 7 and 14, with a stronger enrichment at Day 14 (Appendix A), while the proteasome pathway was enriched in upregulated DA proteins at Day 14 (Figure 4, Appendix A). Within the proteasome pathway, the proteins important for the formation of immunoproteasomes, which play a role in preprocessing antigens for presentation on MHC class I molecules, were upregulated (Figure 4, Appendix A).

Among immunologic signature gene sets, 5 and 6 signatures were enriched in DA results from LC-MS/MS and 2D-DIGE/MS data, at Day 7 and 14, respectively (Table 3 and Appendix A). Both *F. tularensis and A. phagocytophilum* are intracellular gram-negative bacteria, which survive and propagate within the host cells. Immunologic signature gene sets related to *A. phagocytophilum* infection were enriched in DA proteins in the data generated from both experiments (Table 3 and Appendix A). At Day 14, 20 protein families were overlapping for the LC/MS/MS data, with the *A. phagocytophilum* infection set, and 8 protein families were overlapping for the 2D-DIGE dataset (Appendix A and Table 4 and Appendix A). These gene sets are based on published human transcriptomic responses in human polymorphonuclear leukocytes following *A. phagocytophilum* infection [26,27]. These latter enrichment results are highly relevant, based on the live-attenuated *F. tularensis* vaccine used for this study.

The peroxisome proliferator-activated receptor gamma (PPARγ) protein is a major player in resolving inflammation. Lipids 5-HETE, OEA and AEA have all been shown to be ligands for the PPARγ. The LC- MS/MS DA results at Day 14 showed enrichment in published PPARγ-related immunologic signature gene sets, including GSE25123 (gene expression signals in macrophage-specific PPARγ knockout mice) [28,29] with 9 overlapping protein families, GSE37532 (gene expression signals in T cells in mice) [30,31] with 7 overlapping protein families, and GSE37532 (gene expression signals in T-cells obtained from visceral adipose tissue in mice) [31,32] with 5 overlapping protein families (Table 4 and Appendix A). The Day 14 2D-DIGE DA data also showed an enrichment in PPARγ-related GSE37532 gene set (gene expression signals in T cells from PPARγ knockout mice) [31,33], with 4 overlapping protein families (Table 4 and Appendix A).

### 3.3. Identification of Protein Responses that Best Predicted Humoral and Cellular Responses

To assess the relationship between proteome changes and humoral and cellular responses post-vaccination, we utilized CD4^+^, CD8^+^cell activation and tularemia-specific microagglutination titer we collected previously [14]. We applied regularized linear regression models and visualized results using correlation networks (Figure 5). At both days, peak CD4^+^ and tularemia-specific microagglutination titer were correlated with changes in proteins. At Day 7, a change in proteasome subunit alpha type-2 (PSMA2), known to be involved in MHC Class I antigen presentation, was increased from pre-vaccination and was associated with the tularemia-specific microagglutination titer. The same was observed for ribonuclease inhibitor 1 (RNH1), a ribonuclease/angiogenin inhibitor potentially involved in redox homeostasis, which was also increased and correlated with microagglutination titer at both Day 7 and 14 (Figure 5). An increase in spectrin alpha 1 (SPTA1), keratin 1 (KRT1), solute carrier family 4, anion exchanger, member 1 (SLC4A1), proteolytic signal containing nuclear protein (PCNP) and coactosin-like F-actin binding protein (COTL1) was positively associated with peak CD4^+^ activation (Figure 5).

## 4. Discussion

This study expands the efforts to comprehensively characterize immune responses to Tularemia vaccination [14,15,23]. Using both 2-DIGE and LC-MS/MS, changes in proteins in 10 subjects following vaccination were assessed. Three non-human proteins were used as spiked-in controls for both methods for the evaluation of the variability between experiments. Spike-in control protein signals and variability metrics for 3 controls showed that both assays had difficulties with quantifying RNaseA levels. We attribute the lower recovery of RNaseA protein/peptides in the LC/MS/MS analysis to the abundance of cysteine residues and low complexity regions, yielding few proteotypic peptides for sequencing [34]. For the 2-DIGE experiment, we hypothesize that the missing signal of RNaseA was related to its high isoelectric point value (pI = 9.6). On the other hand, bovine beta-lactoglobulin and horse myoglobin were detected in, on average, 72% and 82% of samples, and showed similar variability for both methods.

The analysis of non-control proteins confirmed that the two different proteomic assays, as expected, produced complementary results [35,36]. This was evident by the small number of shared DA proteins and weak correlation between protein abundance and fold change metrics. In addition, 41% of DA spot proteins identified using MALDI-QIT-TOF were not identified by LC-MS/MS, with a reasonable sample coverage (i.e., ≥80% non-missing values) indicating that missing protein identifications/false negatives play a substantial role. These findings can be explained by multiple factors. First, the proteomics strategies used are technologically very different [36,37]. 2D-DIGE performs analysis on intact proteins, while shotgun LC-MS/MS uses digested peptides. In 2D-DIGE, a mixture of intact proteins is surface labeled with fluorophores, such that the ratio of sample to control fluorescence intensity is proportional to the change in abundance. The labeled proteins are separated by 2-dimensional electrophoresis, which has relatively high resolving power and fluorescence dynamic range but is limited in the total protein applied to the gel, which favors the recovery of high abundance proteins. In LC-MS/MS methods, a chromatographic separation of tryptic peptides is carried out, concurrent with MS/MS sampling of the eluted peptides at high frequency to extract peptide sequence information. These data were analyzed using MaxQuant software to identify proteins in the original lysate and to obtain relative quantifications [37]. These significant differences in the read outs are very likely to affect which proteins are identified from a complex mixture. Thus, it is not surprising that different results in terms of DA protein overlap and abundance and fold change metrics were seen. Second, the preparation of PBMC cell lysates was carried out differently for the two experiments, which may have contributed to the observed differences. Sample preparation methods prior to the lysis of the cells could affect the results, including thawing temperatures, speed of thawing, and cell viability assessments that were not performed for LC- MS/MS analysis. There may be additional differences between the two experiments that are less obvious, and which could affect outcomes. When comparing the data, a higher level of albumin contamination/background was observed in the LC-MS/MS samples. While the 2D-DIGE method separated the albumin out as a single spot (Spot ID: S0522, Appendix A), for LC-MS/MS, the resulting albumin peptides were oversampled throughout the gradient. This led to fewer identified peptides and lower quantitative signals for the proteins (peptides) that co-elute. This was evidenced by a strong negative correlation between the abundance of albumin protein and the median protein abundance per sample (Pearson Correlation r = −0.88) in the LC-MS/MS data. This implies that albumin contributed to false negatives and lower fold change accuracy in the LC-MS/MS experiment, by pushing other protein abundances below or near the detection limit. The analysis accounted for this systematic abundance effect using median normalization (scaling up abundances for samples with relatively low medians and scaling down abundances for samples with relatively high medians). However, this median-normalization approach is not designed to correct for missing proteins or the inaccuracy of protein abundances that are pushed close to the detection limit, due to albumin oversampling.

Both proteomic assays indicated that processes related to antigen processing and presentation, and *A. phagocytophilum* infection, were enriched in DA proteins at Day 7 and/or Day 14. The antigen presentation signal at Day 14 showed an increase in the abundance of proteins essential for immunoproteasome formation and an increase in the chaperone protein abundance which is important for MHC class I loading. For the latter, the 2D-DIGE results showed an increasing signal between Days 7 and 14 (both in the number of DA proteins involved in this process as well as in fold change, Figure 3), indicating that MHC class I antigen loading increased during that time. It is known that phagosomes will be formed by antigen presenting cells (APCs) to engulf bacterial pathogens. The phagosomal contents will then be processed after phagolysosomal fusion, and the resulting fragments presented to the antigen specific T lymphocytes in the context of major histocompatibility complex (MHC) surface molecules. Phagocytosis plays a traditional role in providing ligands for MHC-II, but recent studies suggest that phagosomes might alter the conventional pathway for MHC-II antigen processing in yet undefined ways. MHC-I molecules are normally loaded with peptides derived from cytosolic proteolysis. Although *F. tularensis* shows an extracellular phase during bacteremia in mice, survival and replication within host cells is thought to rely on physical escape from its original phagosome and replication in the host-cell cytosol. As expected, the data revealed that, in addition to the upregulated phagosome pathway, the MHC class I antigen processing and presentation, and proteins related to MHC class II antigen presentation (KIF5B, APZA1) and CD8^+^ T cell activation (TPI1) were also upregulated post-vaccination. These findings suggest that the LVS is likely to be partially inactivated, and the associated antigens are processed in the phagolysosomes. Processed antigens then enter the MHCI-II antigen presentation pathway, wherein *F. tularensis* antigens surviving in the cytosol of host cells are later processed by the proteasomes for endogenous MHC class I antigen processing.

To further assess these proteomics results in context with the corresponding transcriptomics results based on RNA-Seq (unpublished data), pathway maps were generated that were color-coded by gene expression for KEGG pathways enriched in DA proteins for the complete time course (Days 1,2, 7, and 14) (Appendix A). These results indicated that gene expression within the MHC I sub pathway was significantly up-regulated for PA28 (immunoproteasome subunit), HSP70, and TAP1/2 at Day 2 post-vaccination. These gene expression results returned to near pre-vaccination levels by Day 7. The gene expression of HSP70 remained significantly upregulated from pre-vaccination at Day 7 and Day 14, matching the proteomics results for Day 7 and Day 14. However, in contrast to those proteomic findings, peak levels were observed at Day 2 and increases in gene expression levels for the MHC 1 sub pathway between Days 2 and 14 were not observed. Gene expression signatures related to immunoproteasome formation were up-regulated throughout Days 1–14, but the strongest responses were detected at Day 2 post-vaccination (Appendix A). This indicates that, as expected, peak transcriptomic responses for MHC 1 antigen processing and presentation preceded the corresponding proteomic responses seen at Day 7 (Appendix A) (note, Day 2 samples for proteomics were not tested).

In addition, on the transcriptomics level, there was evidence of an MHC II sub pathway signal, including a significant upregulation of the gene encoding for the MHCII molecule at Day 7 (Appendix A) and the CIITA gene at Day 14 (MHC class II transactivator). While none of the proteins in the MHC II sub pathway showed differential responses at the proteomics level (Appendix A), the KIF5B and APZA1 proteins that are known to play a role in MHC class II antigen presentation were upregulated in the proteomics experiment at Day 14. Together, these results imply a staggered activation with MHC I presentation to CD8 T cells being activated, first followed by MHC II presentation to CD4 T cells being activated second.

Moreover, both proteomic assays also showed that DA proteins were enriched in PPARγ-related gene sets at day 14 (Appendix A, Table 4). PPARγ is a ligand-activated transcription factor of the nuclear receptor superfamily, that controls the expression of a variety of genes involved in fatty acid metabolism [38]. Endogenous ligands of PPARγ include fatty acids and eicosanoids [39]. PPARγ can alter macrophage trafficking, and increases efferocytosis and phagocytosis. It also plays a role in the adaptive immune response, particularly regarding B cells [40]. In addition, the activation of PPARγ can shift production from pro- to anti-inflammatory mediators [40]. Interestingly, lipidomic analyses of plasma samples derived from the same subjects revealed increased metabolism (decreased abundance) of the pro-inflammatory 5-hudroxyeicosatetraenoic acid (5-HETE, an eicosanoid) lipid by Day 7 post-vaccination, associated with an apparent compensatory increase in dihydroxyeicosatetraenoic acid (DHET) lipid levels [23]. The pro-inflammatory function of 5-HETE is known to be regulated through conversion to its inactive and less active metabolites, DHET lipids by the Cytochrome P450F family of proteins. Related transcriptomic analyses demonstrated that gene *CYP4F22* was upregulated on Day 2 post-vaccination (1.7-fold increase), Day 7 post-vaccination (1.5-fold increase), and Day 14 post-vaccination (1.8-fold increase). Several Cytochrome P450 family genes, including *CYP1B1* and *CYP4V2*, were also differentially expressed [23]. Because cytochrome P450 genes are important for the conversion of pro-inflammatory 5-HETE into inactive DHET metabolites, the transcriptomic results further supported the conclusion that 5-HETE is induced early in the acute response to live tularemia vaccination, followed by the induction of Cytochrome P450 gene expression and the subsequent conversion of 5-HETE into inactive DHET species. Consistent with these findings, Muktha et al. found that Tularemia DVC-LVS vaccination induced the production of IFN-gamma detectable in plasma, a signature proinflammatory cytokine, resulting in a statistically significant increase at Day 1 and Day 2 relative to pre-vaccination [15]. Taken together, the data from proteomics, lipidomics and transcriptomics provided interesting new insights into the interplay between Cytochrome P450 family genes, the pro-inflammatory eicosanoid lipids and PPARy in resolving the Tularemia vaccine induced inflammation.

In addition, at least eight identified proteins were positively associated with peak tularemia-specific microagglutination titer or peak CD4^+^ activation following vaccination. To our knowledge, this is the first report linking these proteins to human immune responses except for PMSA2 and COTL1. PMSA2 is a subunit of the immunoproteasome which is known to be able to enhance the repertoire of peptides presented by MHC-I molecules [41]. In line with this observation, an increase in PSME1 and PSME2 (Proteasome activator complex subunit 1 and 2) protein following AS03 adjuvanted H5N1 influenza vaccination was observed in monocytes at Day 3, predicting later seroprotection status (based on protective levels of HAI titter) [42]. Furthermore, COTL1, a member of the actin depolymerizing factor (ADF)/cofilin family, is related to the actin-binding protein coactosin, and has been identified as a potential regulator of T cell activation by Kim J. et al. [43] Whether these proteins can be used as novel biomarkers to predict human immune responses remains to be established in larger studies using well established methods, including ELISA, western blot analysis or multiplexed quantitative assays.

## 5. Conclusions

Overall, the analysis confirmed that the two different proteomics assays, as expected, produced complementary results. Some commonalities between the results were identified at the pathway level, including antigen presentation, phagocytosis, inflammation (PPARγ receptor), and gram-negative bacterial infection (*A. phagocytophilum* infection) signals. These results support the biological conclusion that, 14 days following vaccination with a live attenuated tularemia vaccine, abundance for proteins involved in both MHC class I and class II antigen presentation were increased compared to pre-vaccination, including immunoproteasome formation proteins and chaperones for MHC class I loading. In addition, the immunoproteasome subunit protein PMSA2 was associated with tularemia-specific microagglutination titer, linking the MHC class I-related response to the later peak humoral response. Consistent with the proteomics data, our transcriptomic alterations showed that MHC class I and class II-related signals were up-regulated at the gene expression level in a staggered fashion, most likely related to differential timing of transcription first, followed by new protein production, and then increased antigen presentation activity.

The corroborating and complementary data from three ‘omics technologies (proteomics, lipidomics, and transcriptomics) provide evidence of how inflammatory response pathways are activated and resolved following live attenuated Tularemia vaccination [15,23]. Although this study was based on a small sample size (10 subjects), the observed DA protein and pathway responses provide new biological insights that can be built upon in future studies, using live attenuated Tularemia strains or other LVS as immunogens, including the assessment of their potential as surrogates of protection.

## Figures and Tables

**Figure 1 vaccines-08-00413-f001:**
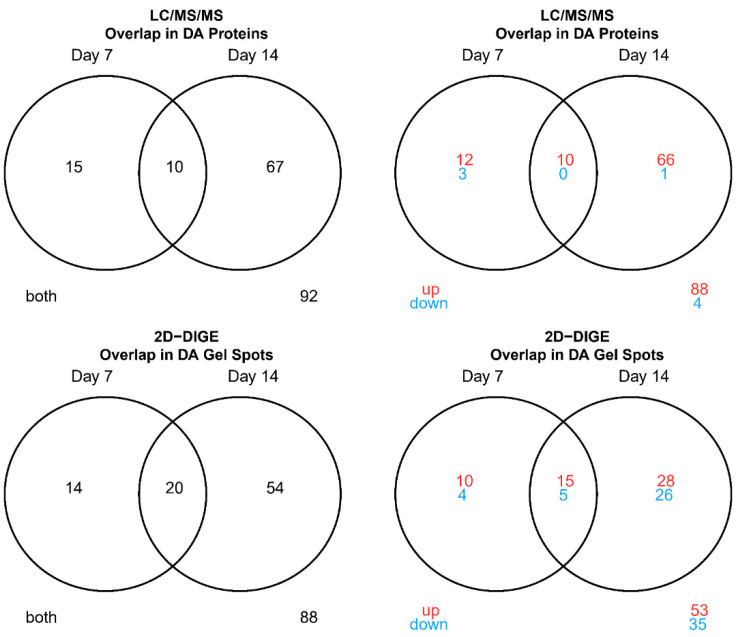
Venn diagrams summarizing overlap in differentially abundant (DA) proteins between post-vaccination days (LC-MS/MS and 2D-DIGE/MS). In red: up-regulated compared to pre-vaccination, in green: down- regulated compared to pre-vaccination.

**Figure 2 vaccines-08-00413-f002:**
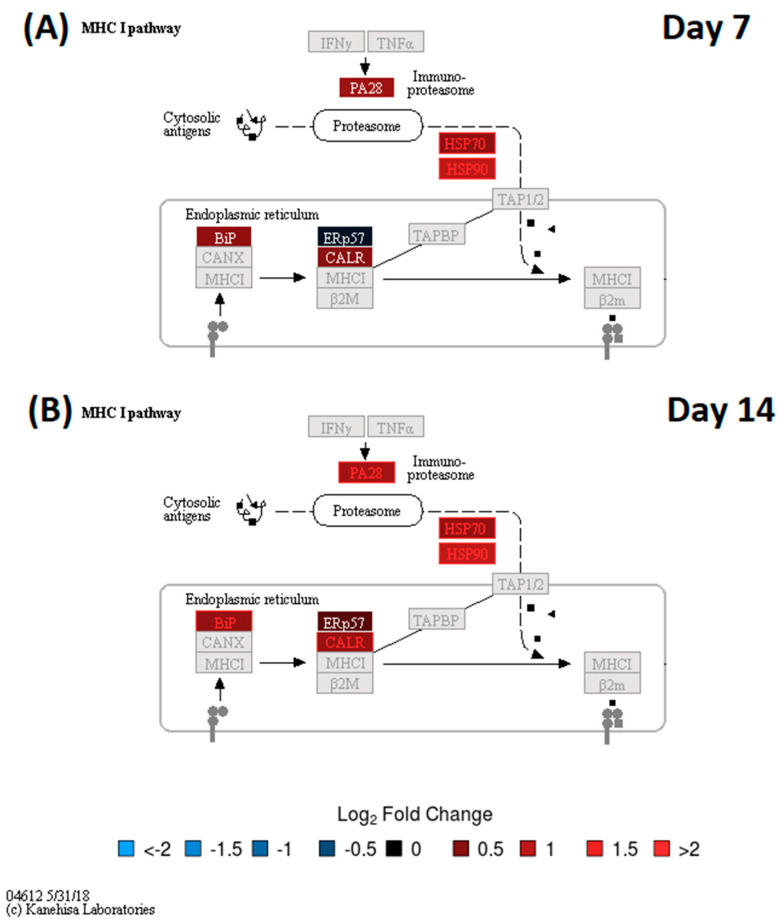
KEGG Pathway Map—Antigen processing and presentation (**A**) Day 7 2D-DIGE results (**B**) Day 14 2D-DIGE results. Node color gradient encodes fold change from pre-vaccination (for multi-gene nodes, the median fold change is used). In red: up-regulated compared to Day 0, in blue: down- regulated compared to Day 0, in black: fold change close to 1, in dark grey: proteins were not identified, light grey: gene missing database mapping. Statistically significant fold changes are highlighted using red and blue label colors.

**Figure 3 vaccines-08-00413-f003:**
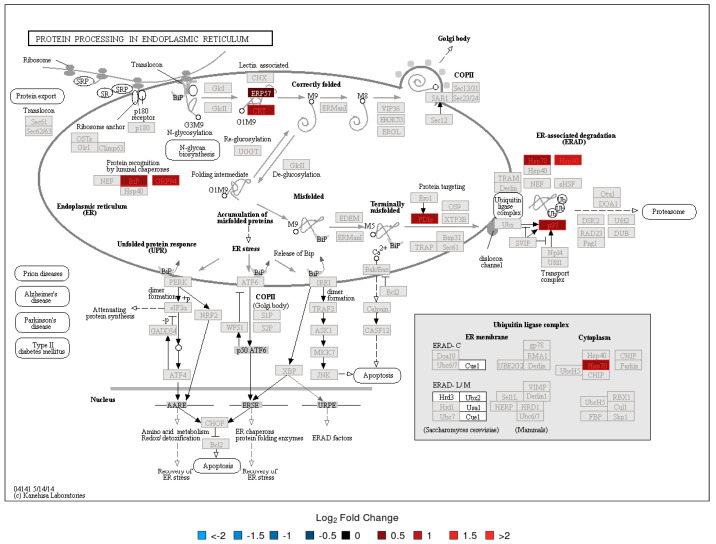
KEGG Pathway Map—Protein processing in endoplasmic reticulum Day 14 2D-DIGE/MS results. See Figure 2 caption for interpretation details.

**Figure 4 vaccines-08-00413-f004:**
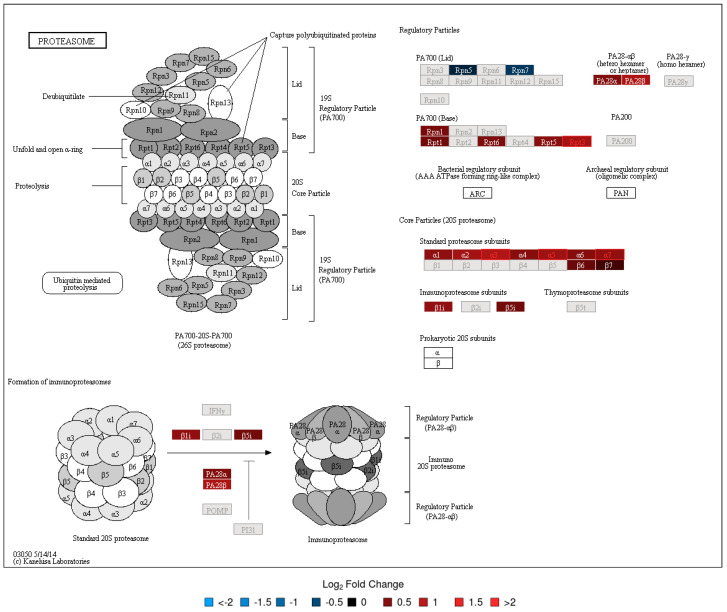
KEGG Pathway Map—Proteasome (LC-MS/MS, Day 14). See Figure 3 caption for interpretation details.

**Figure 5 vaccines-08-00413-f005:**
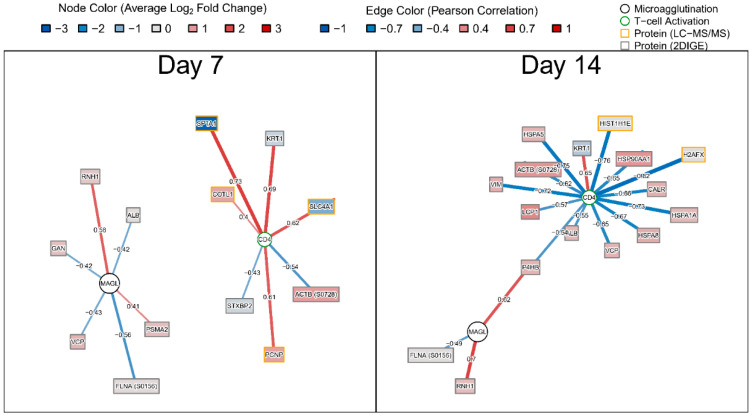
Correlation network summarizing associations between changes in proteins and peak cellular and humoral immune responses. Nodes represent protein fold changes (rectangles), peak CD4^+^ or CD8^+^ T-cell activation across Days 7, 14, or 28 (green circles), or tularemia-specific peak microagglutination titer across Days 14 and 28 (orange circles). Edges represent the Pearson correlation between log_2_ protein fold changes and peak CD4^+^, CD8^+^ percent T-cell activation, or log_2_ microagglutination titer. Protein nodes are color-coded by log_2_ fold change for the respective post-vaccination day. Edges are color coded and edge widths are scaled by Pearson correlation. Nodes were filtered based on Pearson correlation ≥ 0.4 and inclusion as predictors, as part of regularized linear regression models. No correlations were identified for peak CD8^+^ T-cell activation.

**Table 1 vaccines-08-00413-t001:** Overlapping differentially abundant proteins (LC-MS/MS and 2D-DIGE/MS). Protein annotations are based on UniProt annotations (March 16, 2016).

Protein ID	Gene Name	Protein Description	50% Protein Cluster ID (Gene Name)	LC-MS/MSTime Point	2D-DIGE/MS Time Point	LC/MS/MS Log_2_ Fold Change	2D-DIGE Log_2_ Fold Change
P33176	KIF5B	Kinesin-1 heavy chain	Q12840 (KIF5A)	Day 7	Day 7	1.11	0.33
P33176	KIF5B	Kinesin-1 heavy chain	Q12840 (KIF5A)	Day 7	Day 14	1.11	0.35
P07900	HSP90AA1	Heat shock protein HSP 90-alpha	P07900 (HSP90AA1)	Day 14	Day 7	1.12	1.00
P60174	TPI1	Triosephosphate isomerase	P60174 (TPI1)	Day 14	Day 7	1.12	0.58
P07900	HSP90AA1	Heat shock protein HSP 90-alpha	P07900 (HSP90AA1)	Day 14	Day 14	1.12	1.02
P52907	CAPZA1	F-actin-capping protein subunit alpha-1	P52907 (CAPZA1)	Day 14	Day 14	0.56	0.58

**Table 2 vaccines-08-00413-t002:** Enriched KEGG Pathways (LC-MS/MS and 2D-DIGE/MS). Results within assay and day are sorted by false discovery rate and Jaccard similarity coefficient.

Assay	Day	Pathway	Pathway Genes #	Significant Genes (%) [Protein Families]	Genes Up (%)	Genes Down (%)	Jaccard Similarity Coefficient	*p*	FDR-Adjusted P
LC-MS/MS	Day 7	Ribosome	127	5 (3.9) [5]	5 (3.9)	0 (0)	0.036	<0.0001	0.003
	Day 14	Ribosome	127	13 (10.2) [14]	13 (10.2)	0 (0)	0.0798	<0.0001	<0.0001
		Proteasome	44	4 (9.1) [4]	4 (9.1)	0 (0)	0.0449	0.0003	0.0473
2-DIGE/MS	Day 7	Pathogenic Escherichia coli infection	54	3 (5.6) [3]	3 (5.6)	0 (0)	0.2273	0.0001	0.0287
		Antigen processing and presentation	64	3 (4.7) [2]	3 (4.7)	0 (0)	0.046	0.0002	0.0287
		Estrogen signaling pathway	97	3 (3.1) [2]	3 (3.1)	0 (0)	0.0333	0.0006	0.0653
	Day 14	Protein processing in endoplasmic reticulum	159	8 (5) [7]	8 (5)	0 (0)	0.0481	<0.0001	<0.0001
		Antigen processing and presentation	64	5 (7.8) [4]	5 (7.8)	0 (0)	0.0745	<0.0001	0.0002
		Pathogenic Escherichia coli infection	54	4 (7.4) [4]	4 (7.4)	0 (0)	0.2133	<0.0001	0.0023
		Estrogen signaling pathway	97	4 (4.1) [3]	4 (4.1)	0 (0)	0.0388	0.0002	0.0175
		Legionellosis	48	3 (6.2) [2]	3 (6.2)	0 (0)	0.0366	0.0005	0.0279
		Phagosome	142	4 (2.8) [4]	4 (2.8)	0 (0)	0.0982	0.001	0.0498

**Table 3 vaccines-08-00413-t003:** Overlapping enriched MSigDB Immunologic Signature Sets (LC-MS/MS and 2D-DIGE/MS).

		LC-MS/MS			2D-DIGE/MS		
Category Name	Category Genes	Sig. Genes (%) [Protein Fam.]	Jaccard Similarity Coefficient	FDR	Sig. Genes (%) [Protein Fam]	Jaccard Similarity Coefficient	FDR
**Day 7**							
GSE41978 KLRG1 HIGH VS LOW EFFECTOR CD8 TCELL DN	186	5 (2.7) [5]	0.0244	0.0101	3 (1.6) [3]	0.0234	0.0543
GSE41978 ID2 KO VS ID2 KO AND BIM KO KLRG1 LOW EFFECTOR CD8 TCELL DN	190	5 (2.6) [5]	0.0239	0.0101	5 (2.6) [5]	0.0324	0.0006
GSE2405 0H VS 9H A PHAGOCYTOPHILUM STIM NEUTROPHIL DN	192	5 (2.6) [5]	0.0237	0.0101	5 (2.6) [5]	0.0321	0.0006
GSE3982 EOSINOPHIL VS MAST CELL DN	186	4 (2.2) [4]	0.0194	0.0999	3 (1.6) [3]	0.0186	0.0543
GSE2405 0H VS 24H A PHAGOCYTOPHILUM STIM NEUTROPHIL UP	193	4 (2.1) [4]	0.0188	0.0999	3 (1.6) [3]	0.0226	0.0543
**Day 14**							
GSE2405 0H VS 9H A PHAGOCYTOPHILUM STIM NEUTROPHIL DN	192	20 (10.4) [20]	0.0813	<0.0001	8 (4.2) [8]	0.0485	<0.0001
GSE2405 0H VS 24H A PHAGOCYTOPHILUM STIM NEUTROPHIL UP	193	17 (8.8) [17]	0.068	<0.0001	4 (2.1) [4]	0.0302	0.029
GSE41978 ID2 KO VS ID2 KO AND BIM KO KLRG1 LOW EFFECTOR CD8 TCELL DN	190	12 (6.3) [12]	0.0476	<0.0001	5 (2.6) [5]	0.0306	0.0045
GSE22886 NAIVE CD8 TCELL VS DC DN	187	5 (2.7) [5]	0.0195	0.0598	4 (2.1) [4]	0.0264	0.029
GSE29618 BCELL VS MDC DAY7 FLU VACCINE DN	191	5 (2.6) [5]	0.0192	0.0598	5 (2.6) [5]	0.0304	0.0045
GSE23114 PERITONEAL CAVITY B1A BCELL VS SPLEEN BCELL DN	194	5 (2.6) [5]	0.019	0.0601	4 (2.1) [4]	0.0213	0.029

**Table 4 vaccines-08-00413-t004:** Proteins overlapping with enriched *A. phagocytophilum* infection and PPARγ receptor MSigDB Immunologic Signature Sets. (LC-MS/MS and 2D-DIGE/MS, Day 14).

**GSE2405_0H_VS_24H_A_PHAGOCYTOPHILUM_STIM_NEUTROPHIL_UP**			
Assay	**Gene**	**UniProt ID**	**Day 14 Fold Change**
2D DIGE/MS	ACTB;ACTG1	P60709	2.271;2.004
	TUBA1A;TUBA1B;TUBA1C;TUBA3D;TUBA3E;TUBA8	P68363	2.046
	HSPA8	P11142	1.629
	HSPA5	P11021	1.522
LC MS/MS	NPM1	P06748	2.02
	TKT	P29401	1.784
	RPS21	P63220	1.702
	RPL30	P62888	1.638
	RPL4	P36578	1.602
	RPL5	P46777	1.559
	RPL7	P18124	1.555
	RPL22	P35268	1.526
	RPS29	P62273	1.503
	RPL3	P39023	1.495
	RPL17	P18621	1.449
	EEF1B2	P24534	1.433
	RPS18	P62269	1.428
	RPS27	P42677	1.349
	RPL6	Q02878	1.327
	EIF3I	Q13347	1.309
	BRK1	Q8WUW1	1.271
**GSE2405_0H_VS_9H_A_PHAGOCYTOPHILUM_STIM_NEUTROPHIL_DN**			
**Assay**	**Gene**	**UniProt ID**	**Day 14 Fold Change**
2D DIGE/MS	ACTB;ACTG1	P60709	2.271;2.004
	TUBA1A;TUBA1B;TUBA1C;TUBA3D;TUBA3E;TUBA8	P68363	2.046
	HSP90AA1;HSP90AB1	P07900	2.036
	TUBB;TUBB2A;TUBB2B;TUBB3;TUBB4A;TUBB4B	P07437	1.857
	YWHAZ	P63104	1.848
	CAPZB;CEP128;PSME1	P47756	1.674
	HSPA8	P11142	1.629
	VCP	P55072	1.498
LC MS/MS	HSP90AA1	P07900	2.183
	NPM1	P06748	2.02
	EIF4A1	P60842	1.838
	TKT	P29401	1.784
	HMGB1	P09429	1.75
	RPS21	P63220	1.702
	RPL30	P62888	1.638
	RPL4	P36578	1.602
	RPL5	P46777	1.559
	RPL7	P18124	1.555
	CHTOP	Q9Y3Y2	1.539
	RPL22	P35268	1.526
	RPS29	P62273	1.503
	RPL3	P39023	1.495
	RPL17	P18621	1.449
	EEF1B2	P24534	1.433
	RPS18	P62269	1.428
	RPS27	P42677	1.349
	RPL6	Q02878	1.327
	EIF3I	Q13347	1.309
**GSE25123_WT_VS_PPARG_KO_MACROPHAGE_DN**			
**Assay**	**Gene**	**UniProt ID**	**Day 14 Fold Change**
2D DIGE/MS	ARHGDIB	P52566	1.504
LC MS/MS	SYNCRIP	O60506	2.136
	SAMHD1	Q9Y3Z3	2.118
	PSMA4	P25789	1.814
	PTBP1	P26599	1.635
	PSMA5	P28066	1.595
	RPL5	P46777	1.559
	RPL7	P18124	1.555
	EEF1B2	P24534	1.433
	SNX3	O60493	1.247
**GSE37532_TREG_VS_TCONV_CD4_TCELL_FROM_LN_UP**			
2D DIGE/MS	HSP90B1	P14625	1.636
LC MS/MS	ATIC	P31939	2.222
	NOP56	O00567	2.169
	CCT3	P49368	2.018
	HDGF	P51858	1.801
	EEF1D	P29692	1.578
	NOP58	Q9Y2X3	1.525
	SMAP	O00193	1.452
**GSE37532_TREG_VS_TCONV_PPARG_KO_CD4_TCELL_FROM_LN_DN**			
2D DIGE/MS	LCP1	P13796	2.293
	CALR	P27797	1.558
	HSPA5	P11021	1.522
	P4HB	P07237	1.505
**GSE37532_WT_VS_PPARG_KO_VISCERAL_ADIPOSE_TISSUE_TREG_UP**			
2D DIGE/MS	YWHAZ	P63104	1.848
LC MS/MS	HDGF	P51858	1.801
	HMGB2	P26583	1.757
	TCEA1	P23193	1.661
	PPP6C	O00743	1.638

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
