# Peer review of "Proteomic Analysis of Human Immune Responses to Live-Attenuated Tularemia Vaccine"

_vaccines, 2020, doi:10.3390/vaccines8030413_

Round 1

Reviewer 1 Report

I want to thank the authors for their modification of rare comments I given on my first report.

A unique problem remain, please consider using passive format and not active form (our--> this), to ease the reading of this manuscript.

Author Response

Point 1. Please consider to use passive format and not active form (our -> this) to ease the reading of the manuscript.

Response 1. Sentences in active form in the following lines were changed to passive format: Lines 35-36, 41, 46, 84-85, 92, 109-110, 114-115, 313, 356, 445-447, 522, 531-532, 539-540, 545, 584, 619, and 625.

Reviewer 2 Report

In the submitted manuscript "Proteomic analysis of human immune responses to live-attenuated tularemia vaccine", Chang et al presented proteomic analyses following vaccination with live-attenuated tularemia vaccine. The study is interesting and the resubmitted version of the manuscript has been improved.

Minor comments:

  • References 23 and 24 still confusing. If they are not published yet, then is better to remove and mention in the text as “unpublished data or manuscripts in preparation).
  • Mention the full name of F. tularensis at the first lines of the abstract and the introduction then use the abbreviated genus afterword throughout the whole manuscript
  • I invite the authors to go through the manuscript and check any typos.

Author Response

Point 1. References 23 and 24 still confusing, if they are not published yet, then is better to remove and mention in the text as “unpublished data or manuscript in preparation”

Response 1.  Reference 23 was removed and reference 24 was updated. The rest of the reference numbers were updated accordingly. (Please see lines 114, 144-145, 326, 343, 354, 366, 368, 375, 393-394, 396, 399-400, 403, 405, 407, 422-424, 427, 452, 467, 472, 482, 567-570, 574, 596, 599, 601 and 718-720)

Point 2. Mention the full name of F. tularensis at the first lines of the abstract and the introduction then use the abbreviated genus afterward throughout the whole manuscript.

Response 2. The full name of F. tularensis was used in the first line of the abstract and the introduction. The abbreviated genus was used afterward throughout the rest of the manuscript. Please see lines 32, 54, 122, 520 and 528)

Point 3. I invite the authors to go through the manuscript and check the typos.

Response 3. The authors went through the manuscript and checked for typos. Typos were found and corrected in lines 228, 243, 335, 351, 384-385 and 660.

Reviewer 3 Report

The current version of this manuscript looking better. Still it is not clear what is figure S1 (A, B, and C), time points, etc? The authors should include representative pics for all time points. The authors should mention/link all the supplementary data in the main text. 

Author Response

Point 1. The author should include representative pics of all time points.

Response 1. Representative 2D DIGE images of all time points were included in the revised Figure S1. Please see page 19 in the supplementary data file.

Point 2. The authors should mention/link all the supplementary data in the main text.

Response 2. All key supplementary data were linked in the main text. Please see lines 326, 343, 354, 366, 368, 393-394, 396, 399-400, 403 and 405.

This manuscript is a resubmission of an earlier submission. The following is a list of the peer review reports and author responses from that submission.

Round 1

Reviewer 1 Report

With interest, I read the manuscript vaccines-727704. The Authors collect proteomic data from PBMCs obtained from 10 subjects before and 7 or 14 days after vaccination with live-attenuated tularemia vaccine.

I have, however, several serious reservations:

MC1. The authors use PBMCs i.e. a highly heterogeneous population of the cells, which very much limits the interpretative value of the data. Instead of precise information from the certain population of the cells, such approach yields a mix of multiple signals making it difficult to reliably interpret the data. In transcriptomic or epigenetics, it is possible to at least partly avoid potential confounding by cellular heterogeneity in the analyzed samples in studies which are not performed using sorted blood cell populations. If no cell count data are available, a number of bioinformatic tools have been developed that can estimate and correct for cell type heterogeneity in the analyzed samples based on either reference datasets or reference-free statistical algorithms. Are such solutions present in proteomics? Could they be applied here?

MC2. Why the Authors present the other types of omic data, i.e. RNA-Seq based transcriptomics, metabolomics and lipidomics in other manuscripts, instead of integrating them with the current data? This way, we will always see the part of the picture already much limited by usage of the heterogeneous cell population.

Other comments:

OC1. Figure 1 serves technical validation purposes and should be moved to the supplement.

OC2. Resolution is very low and thus the quality as well in some figures.

Reviewer 2 Report

The manuscript titled “Proteomic analysis of human immune responses 2 to live-attenuated tularemia vaccine” is interesting. I recommend this work for publication with major revisions.

Major revisions:

  1. The authors should include at least one representative gel picture of the 2D-DIGE experiment at Day0, Day7, and Day14 after vaccination for the visualization of unique spots.
  2. Authors have observed the upregulation in the protein levels of TPI1, KIF5B, HSP90AA1, and CAPZA1 after vaccination at Day7 and 14 by both 2D-DIGE and LC/MS/MS method. Authors should confirm it by western blot (2D-DIGE lysate can be used for this purpose).
  3. If authors can show early time point proteomic data like 48 hrs or 72 hrs of vaccination, it would be very convincing.
  4. In Figure 3,4,5 and 6, authors should include the tables for the genes with fold change along with cartoon for better understanding.
  5. Authors should check the levels of TNF-α and IFN-γ in serum/plasma which are associated with antigen presentation and inflammation pathway, in order to link the proteomic data.

Minor revisions:

  1. The authors need to correct the color description for figure 3 legend.
  2. In the main text, authors have mentioned about Figure S29 and S30, and Table S19, but in supplementary data, it is not available.
  3. In Figure 6 legend, authors have mentioned about Day 28 data, but it is not shown in the figure.
  4. Why Enriched KEGG Pathways analysis of LC-MS/MS and 2D-DIGE/MS proteomics data showing completely different pathways (Table 2).
  5. Update the references 15, 23 and 24 years or just mention data not shown.

Reviewer 3 Report

In the submitted manuscript " Proteomic analysis of human immune responses to live-attenuated tularemia vaccine", Yie-Hwa Chang et al investigated the protein expression profile following vaccination with live-attenuated tularemia vaccine using two technologies. Although the study is interesting, there are several points that severely weakened the submitted work. for example, the negative control group (PBS injected volunteers) is missing in the study. Several references were cited although they are nor published yet; they should be mentioned in the text as (unpublished data) and not as a referenced work. Examples for this include references 23 and 24. Also some references are presented in a wrong way (e.g. reference 14 is published in vaccine and not vaccines. Reference 15: if the work is for Natrajan et al. 2020, then it has been published in vaccines and not microarray and the first author is not Muktha et al!). Table S19 and Figures S29, S30, S33 mentioned in the text are missing. The resolution of figures is not clear and the overall data need to be presented in a more clear way.

Reviewer 4 Report

The article by Yie-Hwa Chang et al. is very complete, well-written and demonstrated very interesting results from proteomic analysis of human immune responses to vaccination against a bacteria responsible for potentially severe disease, Tularemia.

Very few modifications have to be done before possible publication of the manuscript : 

Line 242: are really H0 and H1 hypotheses the same? Please correct.

Parts 2.6.1 to 2.6.2. : line breaks could be very useful to help the distinction of the title of the paragraph and core of the text

Results. Could the authors perform a sensitivity analysis of the results obtained by both methods? As they clearly (and truely) insist on the difference between them, it could be of interest to see if pathways/structures could be falsy interpreted as "not impacted" due to the negative result of one of the methods. These data could be of interest as exploratory results, I think.

Finally, I want to highlight that the figures are very well-designed and understandable. Thanks to these schemes, that could be used as graphical abstracts to summarize the paper.